# Learning Physics Priors for Deep Reinforcement Learning

## Abstract

While model-based deep reinforcement learning (RL) holds great promise for sample efficiency and generalization, learning an accurate dynamics model is challenging and often requires substantial interactions with the environment. Further, a wide variety of domains have dynamics that share common foundations like the laws of physics, which are rarely exploited by these algorithms. Humans often acquire such *physics priors* that allow us to easily adapt to the dynamics of any environment. In this work, we propose an approach to learn such physics priors and incorporate them into an RL agent. Our method involves pre-training a frame predictor on raw videos and then using it to initialize the dynamics prediction model on a target task. Our prediction model, SpatialNet, is designed to implicitly capture localized physical phenomena and interactions. We show the value of incorporating this prior through empirical experiments on two different domains – a newly created PhysWorld and games from the Atari benchmark, outperforming competitive approaches and demonstrating effective transfer learning.

## 1 Introduction

Recent advances in deep reinforcement learning (RL) have largely relied on model-free approaches, demonstrating strong performance on a variety of domains (Silver et al., 2016; Mnih et al., 2013; Kempka et al., 2016; Zhang et al., 2018b). Unfortunately, model-free techniques do not have very good sample efficiency (Sutton, 1990) and are difficult to adapt to new tasks or domains (Nichol et al., 2018). This is mainly because a single value function represents both the agent's policy and its knowledge of environment dynamics. On the other hand, decoupling the dynamics model from the policy, possible with model-based RL, allows for better generalization and transfer (Zhang et al., 2018a). However, estimating an accurate dynamics model of the environment while simultaneously using it to learn a policy is challenging and expensive, often leading to sub-optimal policies and slower learning. One way to facilitate this process would be to initialize dynamics models with generic priors for more efficient and stable model-based reinforcement learning.

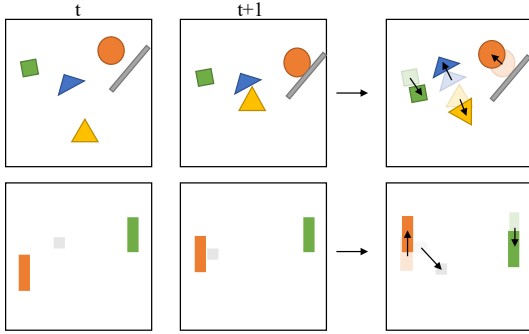

Figure 1: Two different environments with object dynamics that obey the common laws of physics. Agents that have a knowledge of general physics will be able to adapt quickly to either environment.

Consider the scenarios in Figure 1 for example. Both environments contain a variety of objects with different dynamics. For an agent to obtain a good understanding of world dynamics and understand physical laws like conservation of momentum, it has to observe a large number of transitions. For instance, observing just a couple of transitions, the agent cannot infer that the *orange circle* is a freely moving object while the *grey rectangle* is stationary. Further, it would require a significant number of collisions between the circle and the rectangle (at various angles and velocities) to understand the laws governing elastic collisions between two bodies, which is crucial to learning a good model of this environment. Moreover, this entire model learning process has to be repeated from scratch for each new environment. Humans, on the other hand, have reliable priors that allow us to learn the dynamics of new environments very quickly (Dubey et al., 2018). In this work, we demonstrate that learning

a prior of general physics (ex. concepts of velocity, mass, acceleration) allows for better and more efficient estimation of the dynamics of new environments, thereby resulting in better control policies.

In order to learn a prior for physical dynamics, we utilize unsupervised learning over raw videos containing moving objects. In particular, we first train a dynamics model to predict the next frame given the previous k frames, over a wide variety of scenarios with moving objects. This training allows the parameters of the dynamics model to capture the general laws of physics which are useful in predicting entity movements. We can then use this pre-trained dynamics model as a prior in a model-based RL algorithm for a control task. Specifically, we initialize the dynamics model with the pre-trained parameters and finetune them using transitions from the specific task. We utilize this dynamics model in order to predict future frames up to a finite horizon, which are then used as an additional context input into a policy network, similar to the approach of Weber et al. (2017). We show that this results in both faster learning and more optimal policies.

We characterize our dynamics model as an image pixel prediction model. Learning a good future frame model is challenging (Mathieu et al., 2015), mainly because of two reasons - a) the high-dimensionality of the output space with arbitrary number of moving objects and interactions, and b) the partial observability in environments. Existing techniques that use a combination of recurrent and convolutional networks (Oh et al., 2015) suffer from the problem of error compounding and poor generalization as they encode a image into a single vector. In contrast, we propose SpatialNet, a neural network that consists of a convolutional encoder, a spatial memory block, and a convolutional decoder that captures the local nature of dynamics. The spatial memory module operates by performing convolution operations over a temporal 3-dimensional state representation that keeps spatial information intact, similar to the model of Xingjian et al. (2015). This allows the network to capture localized physics of objects such as directional movements and collisions in a more fine-grained manner, resulting in lower prediction error, better generalization and input size invariance.

We evaluate our approach and compare it with existing techniques on two different RL scenarios. First, we consider PhysWorld, a suite of randomized physics-focused games, where learning object movement is crucial to a successful policy. Second, we also evaluate on a stochastic variant of the popular ALE framework consisting of Atari games Bellemare et al. (2013). In both scenarios, we first demonstrate the value of learning a physics prior for model dynamics — our agent achieves up to 130% higher performance on a shooting game, PhysShooter and 49.8% higher on the Atari game of Asteroids, compared to the most competitive baseline. Further, we also show that the dynamics model fine-tuned on these tasks can transfer to new tasks effectively. For instance, our model achieves a relative score improvement of 37.4% on transfer from PhysForage to PhysShooter, than compared to a relative score improvement of 5.4% from a *policy-transfer* baseline.

## 2 RELATED WORK

**Video prediction and reinforcement learning** Our frame prediction model is closest in spirit to the ConvolutionalLSTM which has been applied to several domains including weather forecasting (Xingjian et al., 2015), gesture recognition (Zhu et al., 2017), and forecasting passenger demand (Ke et al., 2017). Similar architectures that incorporate differentiable memory modules have been proposed (Patraucean et al., 2015), with applications to deep RL (Parisotto and Salakhutdinov, 2017). We use a simpler architecture, which we show generalizes better at capturing dynamics, that captures entity movements more directly and demonstrate its use in learning useful physics priors for environment dynamics.

Several recent methods have combined policy learning with future frame prediction in various ways. Action-conditioned frame prediction (Oh et al., 2015; Finn et al., 2016; Weber et al., 2017) has been used to simulate trajectories for policy learning. Predicted frames have also been used to incentivize exploration in agents, via hashing (Yin et al., 2017) or using the prediction error to provide intrinsic rewards (Pathak et al., 2017). The main departure of our work from these papers are that we use a frame prediction model that is not conditioned on actions, which allows us to learn the model from videos independent of the task, and then use it as a prior for learning the specific dynamics of different tasks.

**Physics models for reinforcement learning** Incorporating physics priors into learning dynamics models of environments has been the subject of some recent work. Cutler et al. (2014); Cutler and How (2015) learn Bayesian nonparametric priors for physical systems and use it to guide policy search. Scholz et al. (2014) model latent physical parameters like mass and friction and use an

external physics simulator to predict future states for planning. Kansky et al. (2017) learn a generative physics simulator and demonstrate its usefulness in transferring policies across task variations, but require objects to be detected from raw images. Xie et al. (2016) develop a model that incorporates prior knowledge about the dynamics of rigidbody systems in order to perform model-predictive control. Nguyen-Tuong and Peters (2010) use parametric physics models as priors for learning environment dynamics. While all these approaches demonstrate the importance of having relevant priors to sample efficient model learning, they all require some manual parameterization. In contrast, we learn physics priors in the form of a predictive neural network, directly from observing raw videos.

**Decoupling dynamics from policy** Our work also falls into the general direction of decoupling the agent's knowledge of the environment dynamics from its task-oriented policy. Successor representations Dayan (1993) decompose the agent's value function into a feature-based state representation and a reward projection operator, resulting in better exploration of the state space Kulkarni et al. (2016); Barreto et al. (2017); Machado et al. (2017b). While these state abstractions help with exploration, such representations do not explicitly capture dynamics models of the environment. Zhang et al. (2018a) recently proposed an approach to learn two separate models for dynamics and rewards and use it to perform online planning. However, they learn a dynamics model using task-specific transitions, while we learn a prior from task-independent videos and demonstrate its usefulness in learning different environment dynamics.

## 3 FRAMEWORK

Our goal is to demonstrate that agents can learn useful physics priors from raw videos and employ them effectively to learn dynamics of new environments. To this end, our approach has two phases:

1. *Pre-training*: we first design a suitable neural network architecture (SpatialNet) to predict pixels in the next frame given the previous $k$ frames of a video. We train this network on a newly created video dataset that captures various physical phenomena between entities.

2. *Reinforcement learning*: We use the pre-trained dynamics predictor from the previous phase to initialize the dynamics model for a reinforcement learning agent. This dynamics model is used by the agent to predict a few frames into the future and use them as additional context for its policy. The dynamics model is simultaneously fine-tuned using trajectories observed in the task environment.

We first describe SpatialNet, and then demonstrate how we use it for reinforcement learning.

### 3.1 SPATIALNET

Predicting the physical behavior of an entity requires a model that can perform two crucial operations – 1) isolation of dynamics of an entity, and 2) accurately model localized spaces and interactions around the entity. LSTM-based recurrent networks used in prior work (Oh et al., 2015) are ill-suited for this task since they encode the entire scene into a single latent vector, thereby losing the localized spatio-temporal correlations that are important for making accurate physical predictions. On the other hand, the ConvLSTM (Xingjian et al., 2015) architecture has localized spatio-temporal correlations, but is not able to accurately maintain dynamics of entities due to LSTM state updates and limited separation of stationary and non-stationary objects. In order to satisfy both the above desiderata, SpatialNet uses a spatial memory that explicitly encodes dynamics that are updated with object movement through convolutions. This allows us to implicitly capture and maintain localized physics, such as entity velocities and collisions between entities, in our frame prediction model and allows significantly lower long term prediction error.

**Architecture** SpatialNet is conceptually simple and consists of three modules. The first module is a standard convolutional encoder $\mathcal{E}$ that converts an input image $x$ into a 3D representational map $z$. The second module is a spatial memory block, $\sigma$, that converts $z$ and the hidden state $h$ from the previous timestep into an output representation $o$ and new hidden state $h'$. Finally, we have a convolutional decoder $\mathcal{D}$ that predicts the next frame $x'$ from $o$. Both the encoder and decoder modules ($\mathcal{E}$ and $\mathcal{D}$) use two convolutional layers each with residual connections.

We implement the spatial memory block $\sigma$ as a 2D convolution operation. The module takes in a previous hidden state $h_t$ and input $z_t$ at timestep $t$, both of shape $k \times n \times n$ where $k$ is the number of channels and $n \times n$ is the dimensionality of the grid. We then perform the following operations:

$$i_t = f(C_e \oplus [h_t; z_t]); \quad u_t = f(C_u \oplus [i_t; h_t]); \quad h_t = f(C_{dyn} \oplus u_t); \quad o_t = f(C_d \oplus [z_t; h_{t+1}])$$

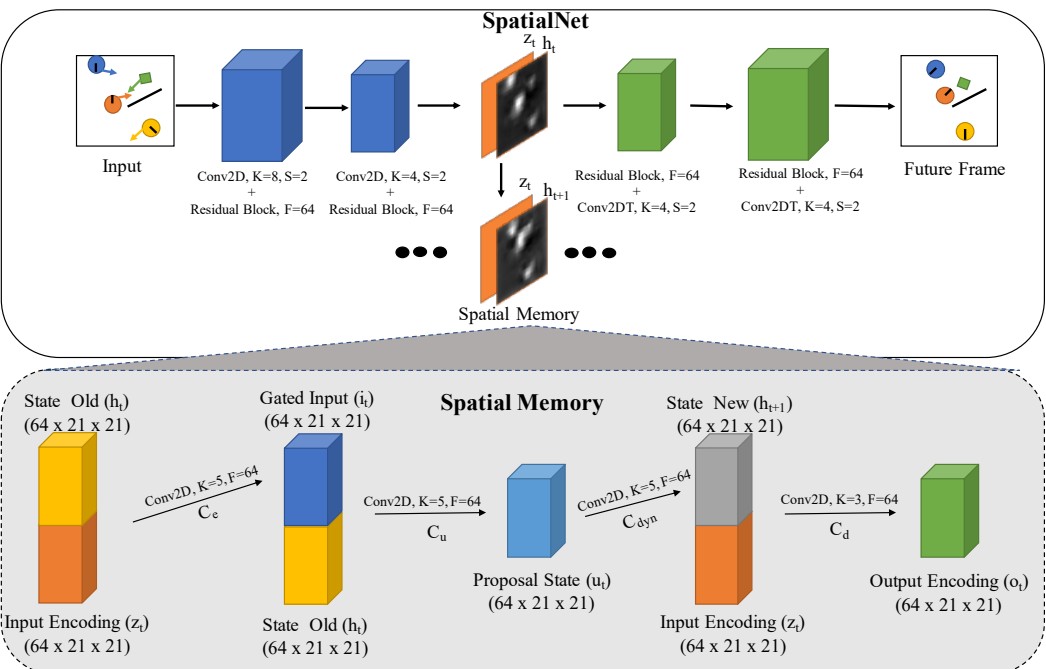

Figure 2: Overview of the SpatialNet architecture. SpatialNet takes an RBG image as input and passes it into encoder ($\mathcal{E}$) consisting of two residual blocks to form an input encoding $z_t$. $z_t$ is processed by a *spatial memory* module ($\sigma$) to obtain an output representation $o_t$, which is used by the decoder ($\mathcal{D}$) to predict the next frame. The spatial memory stores meta information about each entity and its locality. See Section 3 for more details.

where $\oplus$ denotes a convolution operation, $[;]$ denotes concatenation, $C_e$, $C_u$, $C_{dyn}$, $C_d$ are convolutional kernels and $f$ is a non-linearity (we use ELU (Clevert et al., 2015)). The module first encodes a combination of $z_t$ and $h_t$ into a proposal state $u_t$, using two convolutions $C_e, C_u$. $C_{dyn}$ acts like a dynamics simulator and generates a new hidden state $h_{t+1}$, which captures the localized predictions for the next state around each entity. Finally, $C_d$ uses $h_{t+1}$ and $z_t$ to produce $o_t$, encoding information about the entire frame to be rendered by subsequent decoding.

Intuitively, this architecture biases the module towards storing relevant physics information about each entity in a block of pixels at the entity's corresponding location. This information is sequentially updated through the convolutions, while static information such as background texture are passed directly through the input encoding $z_t$. We note that our spatial memory is not action-conditional, which allows us to learn from task-independent videos, as well as generalize better to new environments. Given training videos $D = \left\{ (x_1^{(i)}, x_2^{(i)}, \ldots, x_{T_i}^{(i)}) \right\}_{i=1}^{N}$, we learn the parameters of the model using a standard MSE-based loss function, $L(\theta) = \sum_i \sum_j \|\hat{x}_j^i - x_j^i\|^2$.

## 3.2 REINFORCEMENT LEARNING WITH SPATIALNET

There are several ways one can incorporate a learned dynamics model into a reinforcement learning setup. One approach is to use the model to generate synthetic trajectories and use them in addition to observed transitions while training a policy (Oh et al., 2015; Feinberg et al., 2018). Another direction is to perform rollouts from the current step using the model and then use the predicted states as additional context input to the policy (Weber et al., 2017). Our method is similar to the latter – we use our learned dynamics model to predict $k$ future frames and concatenate these frames along with the current frame to form the input to our policy network. There are two differences however – (1) we predict future state observations *without conditioning on the actions of the agent* and without rewards since our dynamics model is task agnostic, and (2) we do not use a global encoding for future frames, but stack frames and use convolutions to extract local dynamic information.

Formally, we consider a standard Markov Decision Process (MDP) setup represented by the tuple $\langle S, A, T, R \rangle$, where $S$ is the set of all possible state configurations, $A$ is the set of actions available to the agent, $T$ is the transition distribution, and $R$ is the reward function. Assuming our dynamics

model to be $\Omega$, given the current state $s_t$, we apply our prediction model iteratively:

$$\hat{s}_{t+1} = \Omega(s_t), \hat{s}_{t+2} = \Omega(\hat{s}_{t+1}), \ ... \ \hat{s}_{t+k} = \Omega(\hat{s}_{t+k-1})$$

We then train a policy network to output an action using all these predicted states as context in addition to the current state:

$$a_t = \pi(s_t, \hat{s}_{t+1}, \ ... \ \hat{s}_{t+k}) \tag{1}$$

For the policy network, we follow the architecture described in Mnih et al. (2015) and use the Proximal Policy Optimization (PPO) (Schulman et al., 2017) algorithm for learning from rewards obtained in the task. We call this framework an Intuitive Physics Agent (IPA).

We update the policy parameters by using the standard PPO loss, $L(\theta) = \mathbb{E}[\min(r_t(\theta)A_t, \text{clip}(r_t(\theta), 1 - \epsilon, 1 + \epsilon)A_t]$ where $r_t = \frac{\pi_\theta(a_t|\hat{s}_t)}{\pi_{\theta_{\text{old}}}(a_t|\hat{s}_t)}$ and $A_t$ is computed using the value function $V(\hat{s}_t)$. Simultaneously, we also update the parameters of the dynamics model using the same transitions from the environment with the same loss function as in Section 3.1. Policy gradients are not back-propagated to SpatialNet.

## 4 DYNAMICS PREDICTION EXPERIMENTS

We train and evaluate SpatialNet on a variety of 2D environments. These environments require accurate identification of objects and their dynamics for good prediction. To train our model, we generate videos containing moving objects of various shapes and sizes, complete with acceleration, collisions and other physical phenomena.

**Data** We generate a video dataset called *PhysVideos* using the physics engine Pymunk with the included debug renderer (pym). Our dataset consists of frames of size $84 \times 84 \times 3$ containing moving balls and boxes (Figure 3). Each frame has 4-8 random moving circles or squares inside a room with 0-3 randomly generated interior walls. Objects are initialized with friction of 0.9 and elasticity of 0.95, leading to diversity in object movement across a trajectory. All objects are initialized with random positions and velocities. Being able to predict the future in this type of environment requires considerable 2-dimensional physics reasoning, such as inferring velocity from past object movement, anticipating changes in momentum due to collisions, and predicting rotations of each object. We generate 4500 different trajectories for training and 500 trajectories for testing, with each trajectory containing 125 different steps. We train and evaluate the prediction models using the metric of mean squared error (MSE) on a variety of different domains with varying dynamics. We use the Adam optimizer (Kingma and Ba, 2015) in our experiments with a learning rate of $10^{-4}$.

**Baselines** We compare our SpatialNet model with a modified version of the video prediction model of Oh et al. (2015) which we shall refer to as RCNet which uses convolutional and recurrent networks to encode a sequence of frames in order to predict the next frame. While their model also makes use of the agent's action to condition this prediction on, we implement a model without the action-conditioning, i.e. $h_t^{dec} = h_t^{enc}$. We also compare to the ConvLSTM model of (Xingjian et al., 2015) which uses local LSTMs through convolutions where we use kernel sizes of 5 and identical encoders and decoders to SpatialNet.

**Results** From Table 1, we see that SpatialNet acheives a much lower test MSE compared to RCNet and ConvLSTM, especially for multi-step predictions. This suggests that SpatialNet encourages better dynamic generalization compared to RCNet and ConvLSTM. Qualitatively, in Figure 3, we see that SpatialNet is able to maintain the number of objects in video while RCNet suffers from merging of objects. In addition, the predictions of SpatialNet appear much more similar to ground

| Model | 1 step | 3 step | 5 step | 10 step | Objects Lost (20 step) |
|---|---|---|---|---|---|
| RCNet (Oh et al., 2015) | 0.0061 | 0.0099 | 0.0140 | 0.0268 | 1.0 |
| ConvLSTM | 0.0026 | 0.0157 | 0.0303 | 0.0503 | 0.4 |
| SpatialNet | 0.0024 | 0.0076 | 0.0114 | 0.0176 | 0.13 |

Table 1: MSE for multi-step prediction on PhysVideos (test). All models were trained with 1 step prediction loss. SpatialNet suffers less from compound errors during prediction, and is able to maintain objects and dynamics more consistently (Figure 3). Number of objects lost was determined manually by evaluating 15 videos in the test set.

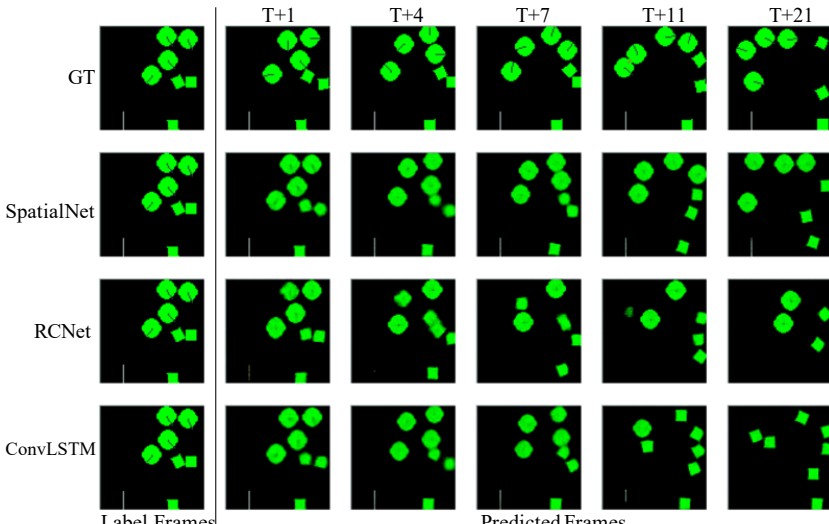

Figure 3: Visualization of multi-step predictions of SpatialNet and RCNet. After 20 steps of self prediction, SpatialNet maintains the internal wall and all seven objects in the scene while RCNet (Oh et al., 2015) loses the internal wall and 3 of the moving objects. ConvLSTM loses shape information and has less accurate dynamics prediction. SpatialNet is the most consistent in obeying physical laws.

truth movement even after 20 steps. Further, SpatialNet is able to maintain background details such as walls that are quickly lost in RCNet, as the spatial memory structure allows the input to easily pass fixed background information. We also find that spatial memory's overall structure allows it to be very resistant to input noise – even when Gaussian noise of magnitude 0.5 is added independently to each pixel in test images, SpatialNet achieves a MSE of 0.0062, while RCNet MSE error rises to 0.0268. We provide a table showing sensitivity to random noise in the appendix.

**Dataset Generalization.** We test generalization by evaluating on two unseen datasets. For the first, we create a test set where objects are half the size of the training set and initialized randomly with approximately twice the starting velocity. In this new regime, we found that RCNet had a MSE of 0.0115, ConvLSTM has a MSE of 0.0067, while SpatialNet had a MSE of 0.0039. We find RCNet is unable to maintain shapes of the smaller objects, sometimes omitting them, while ConvLSTM maintains shape but is unable to adapt to new dynamics. In contrast, SpatialNet local structure allows it to generate new shapes, and its dynamic seperation allows better generalization. In the second dataset, we explore input size invariance. We create a second testing data-set consisting 16-32 random circles and squares and input images of size 168x168x3 (the density of objects per area is conserved). On this dataset, we obtained a MSE of 0.0042 compared to ConvLSTM of 0.0060, which is comparable to the MSE on the original test dataset of 0.0024, showing that the spatial memories local structure allows to easily generalize to different input image sizes. We show qualitative plots of both datasets in the supplementary.

## 5 REINFORCEMENT LEARNING EXPERIMENTS

In this section, we describe the use of SpatialNet to accelerate reinforcement learning. We first train SpatialNet on the physics video dataset described in the previous section. Then, we use the pre-trained SpatialNet model as a future frame predictor for a reinforcement learning agent. We perform empirical evaluations on on two different platforms consisting of 2D games - PhysWorld and a stochastic version of Atari games (Machado et al., 2017a). We demonstrate that IPA with pre-training outperforms other approaches in both platforms. The IPA architecture also allows for effective decoupling of environment dynamics from agent policy, which results in better transfer performance across tasks.[*]

---

[*]Note that the dynamics models we learn do not include prediction of the agent's dynamics (ego-dynamics). This is by design, since ego-dynamics depend on the action space of individual environments and do not easily transfer to new environments.

## 5.1 PHYSWORLD

The first environment we consider is PhysWorld, a new collection of three physics-centric games that we created. These games require an agent to accurately predict object movements and rotations in order to perform well. All three tasks have an environment consisting of around 10 randomly moving boxes and circles as well as up to three internal impassable walls. *PhysGoal* is a navigation task while having to avoid hitting moving objects, *PhysForage* is an object gathering task, and *PhysShooter* requires an agent to shoot selected moving objects while avoiding others. The objects in each of these environments are *different colors and sizes* than those used to train the dynamics predictor in Section 4. We provide a detailed description of each task in the Appendix (A.2). We emphasize that the main parameters (like object velocities, rotations,etc.) in the PhysWorld games are fully **randomized** at the start of each episode. To obtain good performance, agents to really have a good understanding of general physics and not memorize frames.

**Experimental setup.** In our experiments, we use SpatialNet to predict the next k[†] future frames. We then stack the current frame with the k predicted frames and use this as input to a model free policy. We use the Adam optimizer with learning rate 1e-4 to train model predictions and the same set of hyper-parameters for training all policy agents as those used in Schulman et al. (2017). For our policy network, we use the architecture described in Mnih et al. (2015).

**Baselines.** We compare our model with a number of different baselines. The first baseline is a standard implementation of Proximal Policy Gradient (PPO) (Schulman et al., 2017), which is model-free and uses the current frame with the last k frames to output an action. The number of frames provided to PPO is the same as the number provided to IPA.

We also compare with alternative methods to incorporate dynamic models: (1) through value function expansion Feinberg et al. (2018), which uses a dynamics predictor to obtain a more consistent estimate of the current state's value, (2) Imagination Augmented Agents (I2A) Weber et al. (2017) which uses a combination of past frames and a recurrent encoding of future rollouts as input to the policy,[‡] (3) a model that uses the hidden layer of SpatialNet as input to the policy network (ISP), and (4) a model based on ISP but has an auxiliary frame prediction loss (JISP).

Finally, we also consider baselines where we augment the input to PPO with future frames predicted by either RCNet (Oh et al., 2015) or ConvLSTM (Xingjian et al., 2015). We report numbers averaged over 3 different random seeds.

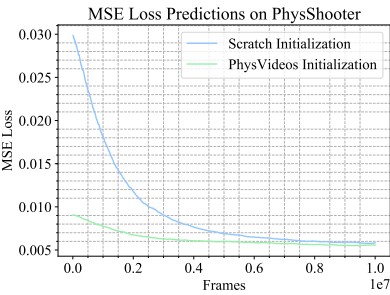

Figure 4: MSE loss when SpatialNet is trained on PhysShooter environment with or without PhysVideos initialization

**Results.** We detail the performance of our approach compared to the baselines in Table 2 and show learning curves in Figure 5. Quantitatively, we find large gains in performance in all three tasks in PhysWorld using IPA with SpatialNet. We find that IPA with RCNet or ConvLSTM provides less benefits, due to slower learning than SpatialNet. We also find PPO with value expansions also provides slight gains, but significantly less than the gains conferred by IPA. We find that I2A leads to no gains in performance, likely due to a global encoding of a image destroying local dynamics information of objects. We also find that both ISP and JISP perform worse than IPA except on PhysForage. On PhysForage, we find that JISP performs better, likely due to increased policy capacity compared to IPA.

IPA encourages the policy to take into account the future physics of objects, a bias crucial for good performance on each of the PhysWorld environments. Qualitatively, we observe that in all three environments, IPA agents navigate to goals and collect objects with more confidence, even if there are nearby obstacles nearby. In PhysShooter, IPA agents are much more able to hit objects further away on the map, which require multiple time-steps before collisions. Figure 4 demonstrates how having a good prior results in faster learning of the environment dynamics of PhysShooter.

---

[†]We find k=3 to work well in our experiments.
[‡]We use the encoding of five future frames predicted by SpatialNet.

| Environment | PPO | PPO + VF | IPA + RCNet | IPA + ConvLSTM | I2A + SpatialNet | ISP | JISP | IPA + SpatialNet (ours) |
|---|---|---|---|---|---|---|---|---|
| PhysGoal | $17.7 \pm 0.1$ | $19.2 \pm 2.4$ | $20.7 \pm 3.1$ | $21.56 \pm 2.1$ | $16.4 \pm 6.2$ | $15.2 \pm 1.2$ | $18.2 \pm 5.5$ | $\mathbf{30.8} \pm 6.2$ |
| PhysForage | $38.9 \pm 8.9$ | $40.4 \pm 5.4$ | $46.3 \pm 23.4$ | $39.47 \pm 7.0$ | $20.75 \pm 2.0$ | $45.3 \pm 5.5$ | $\mathbf{124.3} \pm 27.1$ | $50.6 \pm 11.5$ |
| PhysShooter | $23.2 \pm 1.3$ | $26.1 \pm 2.9$ | $31.7 \pm 1.0$ | $29.1 \pm 1.6$ | $19.3 \pm 0.7$ | $18.6 \pm 1.1$ | $28.6 \pm 1.5$ | $\mathbf{39.1} \pm 2.9$ |

Table 2: Average scores (along with standard deviation) obtained in PhysWorld environments after 10 million frames of training. Scores are rewards over 100 episodes, averaged over runs with 3 different random seeds. IPA + SpatialNet consistently outperforms the other approaches. Both RCNet, SpatialNet, ConvLSTM are pretrained on PhysVideos. PPO+VF = PPO with Value Function Expansion.

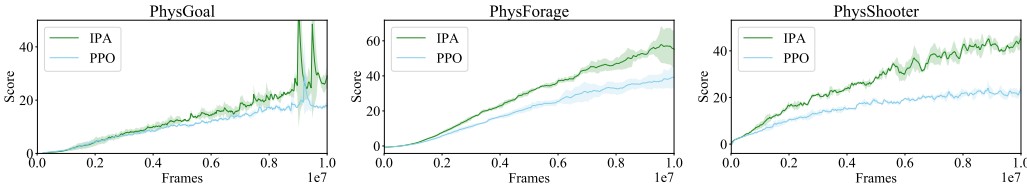

Figure 5: Training curves for PPO and IPA agents on PhysWorld environments. In PhysGoal and PhysForage, IPA demonstrates better performance during later stages of training. In PhysShooter, IPA provides better performance early on because in this game planning is essential since a player can only fire one bullet at a time.

Figure 5 shows the relative training rates of policies under PPO and IPA. In **Phys-Shooter** we see immediate benefits in using a physics model, as physics knowledge of the future is crucial as the agent only gets one action approximately every 4-5 frames. In **Phys-Goal** and **Phys-Forage**, we see long term benefits in knowing future physics as this knowledge allows the agents to more efficiently collect points.

## 5.2 STOCHASTIC ATARI GAMES

In addition to PhysWorld, we also investigate the performance of IPA on a stochastic version of the Arcade Learning Environment (ALE) (Bellemare et al., 2013). The original ALE is fully deterministic (except for random starts) and hence, a dynamics predictor would not provide much value. We modify ALE by adding *sticky actions*, where an agent repeats its last action with probability $p = 0.5$. This stochasticity was shown to be the most challenging type of randomization to add to ALE (Hausknecht and Stone, 2015; Machado et al., 2017a). We evaluate performance on all Atari games and in more detail on selected subset of 10 games though we thought had relevant physics features before evaluation. Experiments on all Atari environments were run with 5 seeds.

**Results.** On overall performance on all Atari games, we find that IPA performs better on 31 games and worse on 18 games as seen in Table 6. In Table 3, we find on our selected games, we observe that IPA outperforms PPO in 8 out of the 10 different tasks. On worse performing tasks, we found no significant degradation. In several games like Enduro, Breakout, Frostbite, FishingDerby and Assault, IPA provides benefits later on in training after the agent has figured out a good initial policy. In others like Asteroids and DemonAttack, IPA provide immediate benefits to training performance, resulting in faster policy learning. One interesting observation is even with the added stochasticity, the dynamics predictor in the Atari domain obtains a very small MSE of around 0.001 or lower, orders of magnitude smaller than the MSE in PhysWorld environments. This indicates that the stochastic Atari environments are still relatively deterministic and simpler in terms of dynamics prediction compared to PhysWorld. We provide additional qualitative results, including frame predictions in the appendix.

## 5.3 TRANSFER AND GENERALIZATION

We now present some empirical results under the transfer scenario and also provide some analysis of our model. Table 4 also shows the impact of initializing IPA with different pre-trained dynamics models on the PhysShooter environment. We find that initializin SpatialNet with random parameters does not perform very well, but using a pretrained SpatialNet pretrained on PhysVideos provides better performance (see Figure 4 for MSE errors). Moreover, we observe that transferring a SpatialNet model fine-tuned on a different task like PhysForage/PhysGoal results in even greater performance improvements. *Interestingly, we note that transferring just the dynamics model in IPA results in a larger performance gains than transferring both model and policy.* For instance, transferring the

| Label | Assault | Asteroids | Breakout | DemonAttack | Enduro |
|-------|---------|-----------|----------|-------------|--------|
| PPO | $2932.2 \pm 153.2$ | $1321.0 \pm 233.5$ | $19.7 \pm 0.9$ | $5510.9 \pm 412.5$ | $376.7 \pm 10.5$ |
| IPA | $\mathbf{2968.4} \pm 124.0$ | $\mathbf{2098.4} \pm 102.0$ | $\mathbf{23.4} \pm 1.0$ | $\mathbf{6793.6} \pm 558.0$ | $\mathbf{398.6} \pm 23.0$ |
| MSE | 0.0023 | 0.0023 | 0.00029 | 0.0032 | 0.00230 |
| Label | FishingDerby | Frostbite | IceHockey | Pong | Tennis |
| PPO | $6.7 \pm 10.1$ | $1342.5 \pm 2154.5$ | $\mathbf{-5.9} \pm 0.3$ | $\mathbf{6.6} \pm 14.1$ | $-6.5 \pm 2.1$ |
| IPA | $\mathbf{9.3} \pm 3.0$ | $\mathbf{1701.1} \pm 2485.0$ | $-6.1 \pm 0.0$ | $2.2 \pm 13.0$ | $\mathbf{-3.8} \pm 1.0$ |
| MSE | 0.00150 | 0.00110 | 0.000035 | 0.00016 | 0.00075 |

Table 3: Scores (and standard deviation) obtained on Stochastic Atari Environments with *sticky actions* (actions repeated with 50% probability at each step). Scores are average performance over 100 episodes after 10M training frames, over 5 different random seeds with included standard deviations.

| Source environment | What is transferred? | Reward | MSE |
|--------------------|----------------------|--------|-----|
| None | PPO | 23.2 | - |
| Random | IPA | 35.42 | 0.00578 |
| PhysVideos | IPA | 39.05 | 0.00554 |
| | PPO | 25.42 | - |
| PhysGoal | Fixed SpatialNet | 26.30 | - |
| | Finetune SpatialNet | **42.83** | 0.00540 |
| | Model + Policy | 42.44 | 0.00540 |
| | PPO | 24.47 | - |
| PhysForage | Fixed SpatialNet | 30.30 | - |
| | Finetune SpatialNet | **53.66** | **0.00533** |
| | Model + Policy | 40.40 | 0.00533 |

Table 4: Effects of model initialization and transfer on training policies in PhysShooter. Topmost section shows baseline PPO, random initialization of dynamics for IPA, and pre-trained IPA using PhysVideos. The bottom two sections demonstrate results while transferring different models from two other games – direct policy (PPO), transfer dynamics model and fix it (Fixed SpatialNet), transfer dynamics and finetune (Finetune SpatialNet), and transfer both dynamics+policy and finetune (Model+Policy). IPA allows decoupling of policy transfer from model transfer, allowing better transfer in cases of environment similarity but task dissimilarity. Scores obtained on the PhysWorld environments after training for 10M frames and evaluated by taking average rewards of the last 100 training episodes.

model from PhysForage results in a score of $53.7$ while transferring both model+policy gets a lower score of $40.4$. This provides further evidence that decoupling model learning from policy learning allows for better generalization.

**SpatialNet Predictions.** Figure 6 shows the qualitative next 3 frame predictions of SpatialNet on each of the different PhysWorld environment with the first frame being the current observation. In PhysGoal, SpatialNet is able to infer the movement of the obstacles, the dark blue agent, and the red goal after agent collection. In PhysGather, SpatialNet is able to infer movement of obstacles as well as the gather of a circle. In PhysShooter, SpatialNet is able to anticipate a collision of the bullet with a moving obstacle and further infer the shooting of a green bullet by the agent.

**Visualization of Spatial Memory.** We provide visualization of the values of spatial memory hidden state while predicting future frames. We visualize the values of spatial memory on PhysVideos, PhysGoal and the Atari environment Demon Attack in Figure 7. To visualize, we take the mean across the channels of each grid pixel in the spatial memory hidden state. We find strong correspondence between high activation regions in the spatial memory and dynamic objects in the associated ground label of the dynamic objects. We further find that static background, such as walls in the input, goals and platforms appear to be passed along in input features.

## 6 CONCLUSION

We have proposed a new approach to model-based reinforcement learning by learning useful physics priors. First, we pre-train a frame prediction model (SpatialNet) on raw videos of a variety of objects in motion. We then use this network to initialize a dynamics model for an RL agent, which makes use

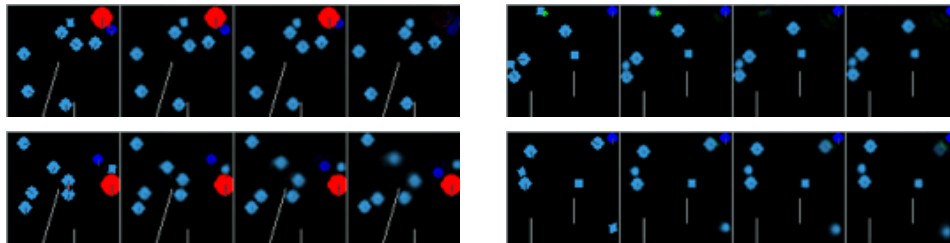

Figure 6: Future image prediction on PhysGoal (left) and PhysShooter (right). First image is current observation, the next three are predicted. SpatialNet is able to predict future dynamics of boxes and balls and anticipate agent movement (PhysGoal) and agent shooting (PhysShooter).

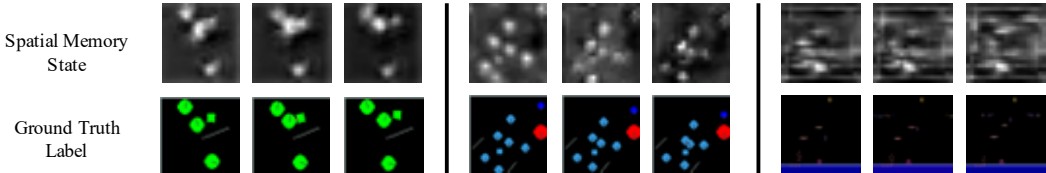

Figure 7: Visualization of SpatialNet hidden state on PhysVideos (left), PhysGoal (middle) and Atari DemonAttack (right). Hidden state has high activations for moving objects while background objects such as walls (left), red goals (middle) and platforms (right) are not attended to as much.

of predicted frames as additional context for its policy. Through several experiments on two different domains, we demonstrate that our approach outperforms model-free techniques and approaches that learn environment dynamics from scratch. We also demonstrate the generalizability of our dynamics predictor through transfer learning experiments.

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

# A APPENDIX

## A.1 ADDITIONAL DYNAMIC PREDICTION EXPERIMENTS

| $\epsilon$ | RCNet | ConvLSTM | SpatialNet (ours) |
|---|---|---|---|
| 0 | 0.0061 | 0.0026 | **0.0024** |
| 0.1 | 0.0078 | 0.0030 | **0.0026** |
| 0.5 | 0.0268 | 0.0072 | **0.0062** |

Table 5: MSE loss on physics prediction data-set on on single-step prediction with test inputs corrupted with Gaussian noise of magnitude $\epsilon$ (model trained with no corruption). Due to its local nature, SpatialNet suffers less form errors in inputs and is able to maintain object numbers/dynamics more consistently even with domain shift.

**Sensitivity to Corruption of Inputs** We investigate the effects of noisy observations in the input domain at test time on both SpatialNet and RCNet, by adding different amounts of Gaussian random noise to input images (Table 5). We find that SpatialNet is more resistant to noise addition. SpatialNet predictions are primarily local, preventing compounding of error from corrupted pixels elsewhere in the image whereas RCNet compresses all pixels into a latent space, where small errors can easily escalate.

**Qualitative visualizations of Generalization Predictions** We provide visualizations of video prediction on each of the generalization datasets in Figure 8 and Figure 9.

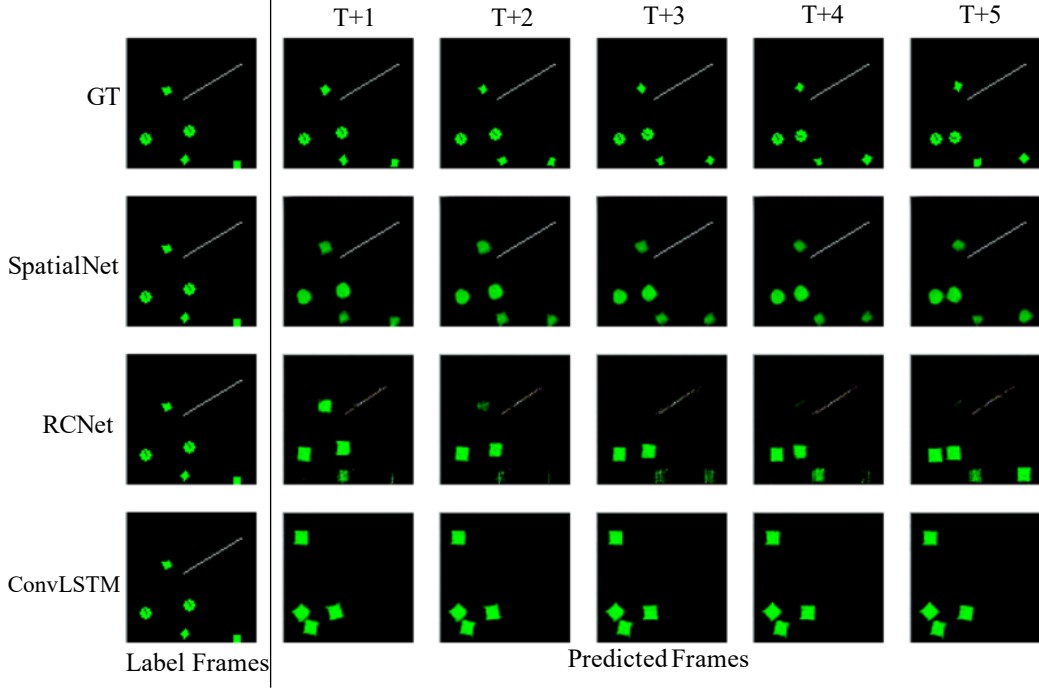

Figure 8: Predictions of SpatialNet, RCNet on test data-set with objects twice as small and with twice the movement speed as trained on. All shown frames are one step predictions. SpatialNet is able to accurately generalize to smaller, faster objects while RCNet is unable to generate the shapes of the smaller objects and suffers from background degradation and ConvLSTM is unable to maintain shapes and dynamics.

## A.2 PHYSWORLD ENVIRONMENTS

We provide a description of the three games environments in PhysWorld:

*PhysGoal:* In this environment, an agent has to navigate to a large red goal. Each successful navigation (+1 reward) respawns the red goal at a random location while collision with balls or boxes terminates the episode (-1 reward).

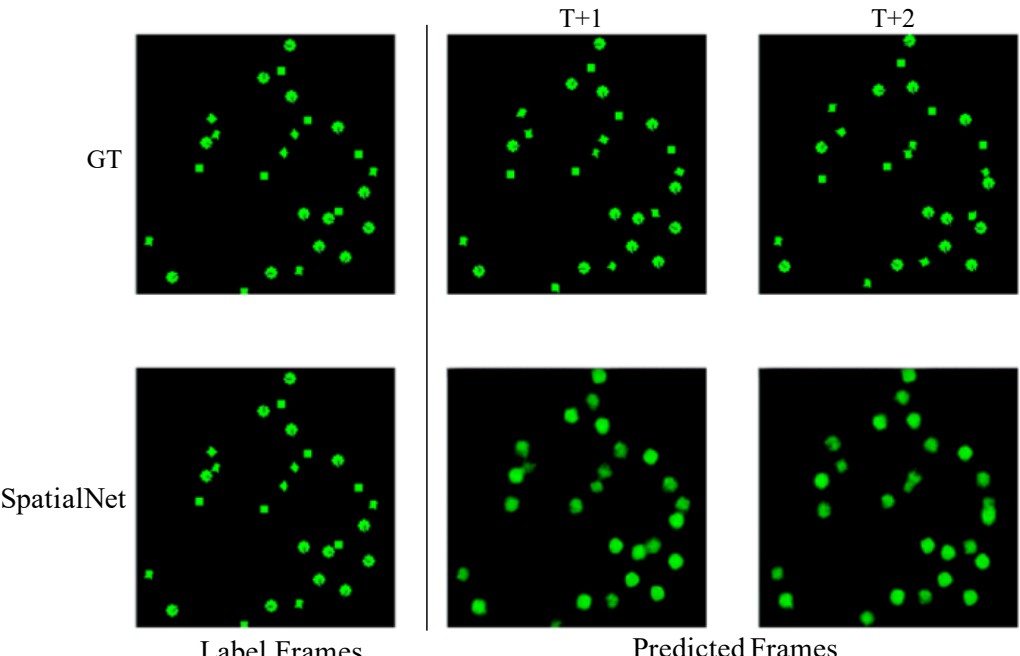

Figure 9: Predictions of SpatialNet on input images of 168 x 168 when SpatialNet was trained on 84 x 84 images. Prediction shown are 1 step future predictions. SpatialNet is able to maintain physical consistency in at large input sizes.

*PhysForage:* Here, an agent has to collect moving balls while avoiding moving boxes. Each collected ball (+1 reward) will randomly respawn at a new location with a new velocity. Collision with boxes lead to termination of episode (-1 reward).

*PhysShooter:* In PhysShooter, the agent is stationary and has to choose an angle to shoot bullets. Each bullet travels through the environment until it hits a square (+1 reward) or circle (-1 reward) or leaves the screen. If a moving ball or box hits the agent (-1 reward), the episode is terminated. After firing a bullet, the agent cannot fire again until the bullet disappears.

Examples of agents playing the PhysWorld environments are given in Figure 10.



Figure 10: Example agent game-play in each of the PhysWorld environments. In PhysGoal (left), the dark blue agent attempts to reach a red goal while avoiding moving objects. In PhysForage (middle), the dark blue agent attempts to gather light blue circles while avoiding squares. In PhysShooter (right), the dark blue agent is immobile and chooses to fire bullet a green bullet at squares while avoiding circles.

A.3   ADDITIONAL REINFORCEMENT LEARNING EXPERIMENTS

**Performance on Atari**   We provide plots of training curves on all Atari environments in Figure 11 on provide on quantitative numbers in Figure 6.

**Predictions on Atari**   We also investigate the benefits (in terms of MSE) of initializing SpatialNet pretrained on the physics dataset compared to training with scratch in Figure 7. We evaluate the MSE error at 1 million frames and find that initializing with the physics dataset provides a 12.9% decrease in MSE error. We find that pretraining helps on 7 of the 10 Atari environments, with

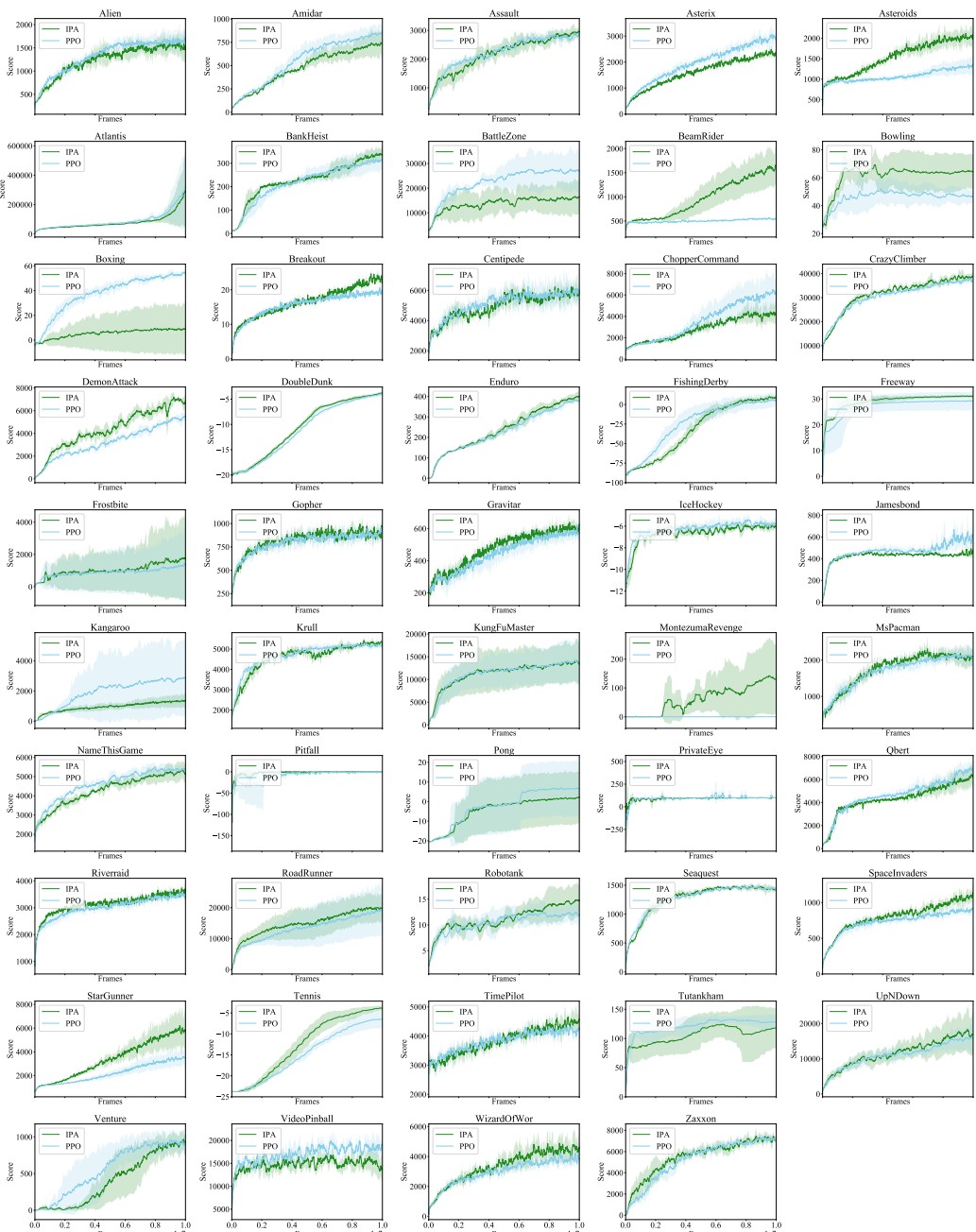

Figure 11: Plots of policy performance trained with either PPO or IPA on all Atari environments on 5 different seeds. IPA sometimes leads to low learning early on the training due to rapid change of 3 predicted future frames. However, later on in training in many different environments, IPA provides performance gains by giving policies future trajectories.

the most negatively impacted environment being Enduro, a 3D racecar environment in which the environmental prior encoded by the physics dataset may be detrimental. More significant gains in transfer may be achievable by using a large online database of 2D YouTube videos which cover even more of diversity of games.

**SpatialNet Predictions** We further visualize qualitative results on SpatialNet on training Atari in Figure 12. In general, across the Atari Suite, we found that SpatialNet is able to accurately model both the environment and agents behavior. In the figure, we seed that SpatialNet is able to accurately

| Environment | PPO | D2A |
|---|---|---|
| Alien | **1668.6** ± 224.3 | 1485.5 ± 281.0 |
| Amidar | **855.9** ± 98.6 | 725.5 ± 135.0 |
| Assault | 2939.2 ± 153.2 | **2968.4** ± 124.0 |
| Asterix | **2920.8** ± 287.3 | 2334.0 ± 184.0 |
| Asteroids | 1321.0 ± 233.5 | **2098.4** ± 102.0 |
| Atlantis | **323205.4** ± 277643.2 | 289369.8 ± 239469.0 |
| BankHeist | 310.4 ± 44.0 | **334.3** ± 29.0 |
| BattleZone | **26828.0** ± 8472.0 | 16526.7 ± 6986.0 |
| BeamRider | 553.1 ± 28.4 | **1630.3** ± 400.0 |
| Bowling | 46.6 ± 5.2 | **64.3** ± 13.0 |
| Boxing | **54.3** ± 2.5 | 8.9 ± 20.0 |
| Breakout | 19.7 ± 0.9 | **23.4** ± 1.0 |
| Centipede | **6043.7** ± 990.6 | 6032.5 ± 199.0 |
| ChopperCommand | **6549.4** ± 1779.1 | 4112.0 ± 1024.0 |
| CrazyClimber | 36893.2 ± 463.9 | **38499.0** ± 1221.0 |
| DemonAttack | 5510.9 ± 412.5 | **6793.6** ± 558.0 |
| DoubleDunk | -4.0 ± 0.5 | **-3.8** ± 0.0 |
| Enduro | 376.7 ± 10.5 | **398.6** ± 23.0 |
| FishingDerby | 6.7 ± 10.1 | **9.3** ± 3.0 |
| Freeway | 29.2 ± 3.6 | **31.2** ± 1.0 |
| Frostbite | 1342.5 ± 2154.5 | **1701.1** ± 2485.0 |
| Gopher | 904.0 ± 42.3 | **941.1** ± 56.0 |
| Gravitar | 574.9 ± 36.2 | **627.2** ± 25.0 |
| IceHockey | **-5.9** ± 0.3 | -6.1 ± 0.0 |
| Jamesbond | **598.9** ± 112.1 | 454.3 ± 34.0 |
| Kangaroo | **2842.4** ± 2461.2 | 1373.0 ± 445.0 |
| Krull | 5178.9 ± 205.1 | **5219.3** ± 129.0 |
| KungFuMaster | **13831.6** ± 4483.6 | 13358.5 ± 4352.0 |
| MontezumaRevenge | 0.0 ± 0.0 | **129.7** ± 122.0 |
| MsPacman | 1990.1 ± 227.9 | **2097.3** ± 259.0 |
| NameThisGame | **5406.4** ± 278.0 | 5131.3 ± 427.0 |
| Pitfall | -0.1 ± 0.3 | **0.0** ± 0.0 |
| Pong | **6.6** ± 14.1 | 2.2 ± 13.0 |
| PrivateEye | 95.6 ± 5.4 | **99.6** ± 0.0 |
| Qbert | **6981.0** ± 548.0 | 6331.4 ± 769.0 |
| Riverraid | 3411.0 ± 201.9 | **3612.4** ± 130.0 |
| RoadRunner | 19329.6 ± 8472.6 | **20041.8** ± 4906.0 |
| Robotank | 11.9 ± 1.8 | **14.9** ± 3.0 |
| Seaquest | **1426.0** ± 43.5 | 1408.7 ± 51.0 |
| SpaceInvaders | 902.4 ± 66.0 | **1132.6** ± 101.0 |
| StarGunner | 3450.0 ± 801.5 | **5778.5** ± 1584.0 |
| Tennis | -6.5 ± 2.1 | **-3.8** ± 1.0 |
| TimePilot | 4281.8 ± 126.6 | **4580.0** ± 314.0 |
| Tutankham | **128.5** ± 12.3 | 118.2 ± 35.0 |
| UpNDown | 15872.3 ± 3995.3 | **16913.7** ± 6344.0 |
| Venture | 930.2 ± 137.9 | **946.7** ± 167.0 |
| VideoPinball | **18878.1** ± 1251.7 | 13981.2 ± 2136.0 |
| WizardOfWor | 3835.6 ± 404.7 | **4629.8** ± 662.0 |
| Zaxxon | 7197.4 ± 220.6 | **7271.0** ± 264.0 |

Table 6: Scores obtained on Stochastic Atari Environments with *sticky actions* (actions repeated with 50% probability at each step). Scores are average performance over 100 episodes after 10M training frames, over 5 different random seeds.

predict agent movement and ice block movement in Frostbite. On DemonAttack, SpatialNet is able to infer the falling of bullets. On Asteroid, SpatialNet is able to infer the movement of asteroids. Finally, on FishingDerby, SpatialNet is able to the right player capturing a fish and also predict that the left player is likely to catch a fish (indicated by the blurriness of the rod). We note that any blurriness in predicted output may in fact even be beneficial to the policy, since policy can learn to interpret the input. We provide training curves and additional analysis on effects of physics transfer on these environments in the supplementary material.

| Environment | MSE PD | MSE DN | Percent Advantage |
|---|---|---|---|
| Assault | 0.00477 | 0.00522 | 9.4% |
| Asteroids | 0.002506 | 0.002518 | 4.7% |
| Breakout | 0.000417 | 0.000423 | 1.4% |
| DemonAttack | 0.00433 | 0.00562 | 29.8% |
| Enduro | 0.00576 | 0.00411 | -28.7% |
| FishingDerby | 0.00183 | 0.00192 | 4.9% |
| Frostbite | 0.000965 | 0.00107 | 10.8% |
| IceHockey | 0.000614 | 0.0013 | 111.7% |
| Pong | 0.00636 | 0.00584 | -8.2% |
| Tennis | 0.00142 | 0.00132 | -7.1% |

Table 7: MSE on Stochastic Atari Environments (a action is repeated with a geometric distribution with p=0.5) at 1 million training frames. MSE PD is trained with a model from physics dataset while MSE DN is trained with a model from scratch. We evaluate percentage advantage for initializing with a physics dataset as compared to from scratch. We average 12.9% decrease in MSE error using a initialization from pretraining on a physics dataset. The most negative environment, Enduro, involves a 3D landscape which initializing from model trained on a physics data set may be detrimental.

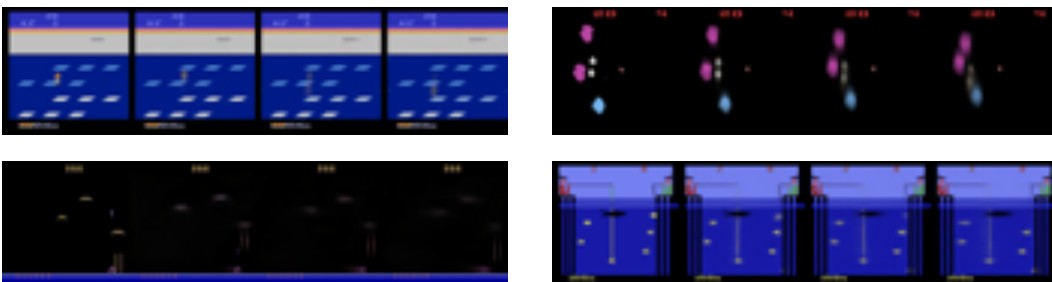

Figure 12: Visualization of model future state prediction on 4 games in Atari (Frostbite - upper left, DemonAttack - lower left, Asteroids - upper right, FishingDerby - lower right). SpatialNet is able to predict falling of bullets, the catching of fish, movement of asteroids, and the movement of tiles/future agent movement in different environments. First frame visualized is ground truth observation, next 3 frames are model future frame predictions.

