# OpenReview forum: "Learning Physics Priors for Deep Reinforcement Learing"
_ICLR.cc/2019/Conference_

### Official Review · AnonReviewer1 · 2018-10-26
**Interesting Direction**

**Rating:** 5
**Confidence:** 5

**Review:**

Quality: The paper proposed a new method to learn some physics prior in an environment along with a new SpatialNetwork Architecture. Instead of learning a specific dynamics model, they propose to learn a dynamics model that is action-free, purely learning the extrinsic dynamics.  They formulate this problem as a video prediction problem. A series of experiments are conducted on PhysWorld (a new physics based simulator) and a subset of Atari games.
Clarity: The writing is good.
Originality: This work is original as most of the model-based RL works are focusing on learning one environment instead of common rules of physics.
significance of this work: This work propose an interesting direction to pursue.

cons:
1. In Figure 4, the authors show that a pretrained model can learn faster than random initialization. However, it is hard to ablate the factor that causes this effect.  Does the dynamics predictor learn the physics priors or is it just because it learn the visual prior of the shape of the objects, etc?
2. The baseline for atari games is quite limited. First of all, 3 out of 5  atari games  in the original PPO paper show that ACER performs better than PPO. (asteroid, breakout, DemonAttack). I think it is better to make some improvement upon state-of-the-art methods.
3. All the experiments are shown with only 3 random seeds, without error bar in the main paper. Although the reward plots are shown in Figure 11.
4. 5 out of 10 atari games are similar to PPO (according to Figure 11). It's hard to be conclusive when half of the experiments are positive and the rest are not.
5. Lack of discussion about ego-dynamics. There are physics priors for both the environment and the controller. Usually the controller/agent  requires an action to predict its dynamics. Then why should we omit the ego-dynamics and only model the outer world.
6. Physics prior usually happen in physical environment. The proposed method works well in the physworld environments. But is there some task that are more realistic than atari games that can leverage the power of physics priors more? It's good that this method works in some atari games. But isn't learning the dynamics of atari games a bit off the topic?
7. The transfer learning experiments should contain a baseline -- maml/reptile. Since you are learning physics prior, it is fair to add meta-learning baselines for comparison.

I think the direction is interesting and the effort is made well. But the experiments are less convincing than the abstract/introduction.

---

> ### Author Response · Authors · 2018-11-19
> **Author Response**
>
> We thank Reviewer 1 for the helpful feedback. We provide answers to individual questions below.
>
> (1) We note that the environments in PhysWorld contain objects of different color and shapes than the objects in PhysVideos, the video dataset used for pre-training SpatialNet. As a result, the pretrained model’s notion of shape or color does not have any transferability to the new task, only its knowledge of physical dynamics does.
>
> (2) When comparing average performance across all the Atari benchmark games, PPO is the state of the art approach, competitive with ACER and better than A2C [Schulman et al., 2017].
>
> (3) We have included standard deviation values for rewards in in PhysWorld and Atari environments (Table 2,3,6) . Following prior work [Schulman et al., 2017], we use 3 seeds for all our experiments. We also ran extra experiments for the Atari games (5 seeds) and observed similar mean and standard deviation performance (Tables 3 and 6)
>
> (4) We have added in results for the entire suite of Atari games in Appendix A.3 (Table 6). Across all the 49 games, IPA outperforms PPO in 31 games. Not all Atari games require an understanding of physical dynamics, which explains why IPA does not improve upon PPO in those games. We specifically use PhysWorld for this purpose -- to test our approach on environments that rely more on understanding basics physics like velocity, collision laws, etc.
>
> (5) Thank you for this suggestion -- we have added a discussion about ego-dynamics in the paper. Since our approach is to learn physics priors that transfer well to new environments, we don’t learn ego-dynamics, which require the action space of the agent to be input to the model -- this is usually task-specific. The dynamics of the world minus the ego-dynamics is more general and transfers well to new environments. See our comparison for transfer with a “model+policy transfer” baseline in Table 4.
>
> (6) We agree that achieving performance gains on many Atari games are limited by factors other physics such as exploration or reflexive action, which we note maybe the reason we do not achieve universal improvement across all Atari games. However, we believe certain Atari games, such as Asteroids, do benefit from predicting the dynamics of moving rocks, etc., and we do observe substantial gains in such environments. We specifically created PhysWorld and performed empirical studies to test our approach on environments that rely more on understanding various aspects of basic physics like velocity, collision laws, etc.
>
> (7) Our transfer learning experiments (Table 4) test the generalization of a policy from a single source environment to a single target environment. In this scenario, techniques like MAML are not directly applicable since they require meta-learning over multiple different environments to find good initialization points for the policy parameters. We also note that methods like PPO do perform better than approaches like MAML on tasks like the Sonic Benchmark [Nichol et al., 2018].
>
>
> References:
> [Machado et al., 2017] Revisiting the Arcade Learning Environment: Evaluation Protocols and Open Problems for General Agents
> [Schulman et al., 2017] Schulman, J., Wolski, F., Dhariwal, P., Radford, A., & Klimov, O. (2017). Proximal policy optimization algorithms. arXiv preprint arXiv:1707.06347.
> [Nichol et al., 2018] Nichol, A., Pfau, V., Hesse, C., Klimov, O., & Schulman, J. (2018). Gotta Learn Fast: A New Benchmark for Generalization in RL. arXiv preprint arXiv:1804.03720.

---

> > ### Comment · AnonReviewer1 · 2018-12-11
> > **response**
> >
> > Thank you for your response. The extra experiments definitely make this paper much convincing. However, learning *physics* priors is not clarified in the text/experiments. The ablation study is not convincing to me to show *physics priors* is different or superior to "imagination augmented"(video prediction) methods. I think the general direction of this paper is good, but the paper needs major revision. Therefore, I am unable to recommend this paper for acceptance.

---

> ### Author Response · Authors · 2018-12-09
> **Request for feedback**
>
> Dear reviewer,
> Thank you so much for your original comments. We spent a large amount of work adjusting the clarifications initially requested. We would appreciate it if you could take a look at the revised version and let us know your thoughts.

---

### Official Review · AnonReviewer2 · 2018-11-02
**Comparison with closely related method is necessary**

**Rating:** 4
**Confidence:** 3

**Review:**

Summary
This paper propose to learning a dynamics model with future prediction in video and using it for reinforcement learning.
The dynamics models is a variants of convolution LSTM and it is trained mean squared error in the future frame.
The way of using dynamics model for reinforcement learning is similar to Weber et al., 2017, where K step prediction of the dynamics model is uses as an augmented input of the policy.

Strength
Training dynamic model to understand physic and using it for reinforcement learning is an interesting problem that worth exploring. This paper tackles this problem and demonstrated experimental setting based on physics games.

Weakness
The part for understanding dynamics model is very close to existing convolutional LSTM model (Xingjian et al., 2015), which is a popular baseline in video modelling community and how pretrained dynamics model is used for reinforcement learning is similar to Weber et al., 2017, but this paper does not provide comparison to any of these two baseline.
Since the difference with these existing method is subtle, clear comparison with these method and difference in characteristic is essential to show the novelty of the paper.

Overall comment
This paper address the interesting problem of understanding dynamics for solving reinforcement learning, but the suggested method is not novel and comparison with existing close methods are not performed.

---

> ### Author Response · Authors · 2018-11-19
> **Author Response**
>
> Thank you for the helpful feedback. We have added relevant comparisons to both the imagination augmented agents architecture (I2A) (Table 3) and the ConvLSTM model (Table 1) as baselines. Below, we provide a comparison of our method with each baseline along with empirical results.
>
> ConvLSTM: SpatialNet differs from ConvLSTM in two main ways that allow it to maintain dynamics information more accurately: 1)  the grid states are updated through convolutions instead of LSTM updates (which blur dynamics over time), and (2)  SpatialNet also has a input copy mechanism that add the current state to the output of the spatialnet encoding -- this allows the encoding to focus better on the dynamics.
>
> We trained a ConvLSTM following the specifications in (Xingjian et al., 2015) and also performed some hyperparameter tuning. We find that the ConvLSTM architecture allows for similar 1 step future frame prediction as SpatialNet (our model) -- see Table 1. However, we find that ConvLSTM  is unable to maintain dynamics information over longer horizons and achieves significantly worse multi step future frame prediction (Table 1 and Figure 3). ConvLSTM also does not generalize well to new datasets with smaller and faster objects (Figure 8). SpatialNet, on the other hand, has a much simpler mechanism for capturing state transitions and is able to effectively model physics and generalize better.
>
> I2A: I2A encodes a global context summary of future frames which is fed into a policy while IPA stacks future frames, allowing convolutional filters to encode local dynamics of different objects.
>
> We trained the I2A model following the specifications in [Weber et al., 2017] and also performed some hyperparameter tuning, where SpatialNet is used as a future frame predictor.
> In our experiments, we find that I2A performs significantly worse than IPA (our approach) and performs on par with PPO on the PhysWorld environments. By feeding stacked future frames in IPA, we allow convolutions to locally extract information about each individual object to predict its dynamics in the future. In contrast, I2A’s structure only allows global encoding of the future states of objects that makes it difficult for policy to infer the future dynamics of objects and their interactions.
>
> References:
> [Weber et al., 2017] Imagination-Augmented Agents for Deep Reinforcement Learning

---

> > ### Comment · AnonReviewer2 · 2018-12-05
> > **Thank you for additional control experiments.**
> >
> > Thank you authors for adding appropriated baselines and comparisons as requested.
> > I increased rating, but I'm still concerned with the focus of the paper.
> > In terms of the writing, it seems the paper's focus is about studying "physics priors for reinforcement learning" in general.
> > However, the title "physics priors for reinforcement learning" seems too general to differentiate the paper from few existing works related to this topic.
> > Rather, I suggest focusing on the difference with existing prior works and put more emphasis on novel contributions would make the paper more clear and easy to understand.
> > Because this requires major revision in the paper, I would keep my rating as rejection for this submission.

---

> > > ### Author Response · Authors · 2018-12-09
> > > **Clarifications**
> > >
> > > Thank you for getting back to us! We explicitly state in the abstract that we propose a method for learning physics priors from raw videos and demonstrate its applicability to deep RL. We have provided detailed comparisons to the most related pieces of work in both the Related Work (Section 2), where we mention the key differences between our work and the prior work, as well as in the experiments sections (Sections 4 and 5), where we provide empirical comparison to these methods. These include the papers you mentioned earlier - ConvLSTM (Xingjian et al.), I2A architecture (Weber et al.) and RCNet (Oh et al.), in addition to other baselines.
> > >
> > > Regarding your comment “Rather, I suggest focusing on the difference with existing prior works” - we would appreciate if you could provide us with any specific papers you had in mind. If you feel the title is too generic, we would also appreciate any suggestions on specific changes.

---

### Official Review · AnonReviewer3 · 2018-11-06
**Interesting Idea, Unclear Writing**

**Rating:** 5
**Confidence:** 4

**Review:**

A method for learning physics priors is proposed for faster learning and better transfer learning. The key idea in learning physics priors using spatial net, which is similar to a convolutional LSTM model for making predictions. Authors propose to improve the sample efficiency of Deep RL algorithms, by augmenting PPO’s state input with 3 future frames predicted by the physics prior model.

Authors show that using Spatial-Net leads to better prediction of the future as compared to previous methods on simple simulated physics environment and can be incorporated to improve performance on ATARI games.

(a) I am a bit unclear on how Spatial-Net is trained along with the policy in the IPA architecture. In section 5.1 it is mentioned that, “We train both SpatialNet and the policy simultaneously and use Proximal Policy optimization (PPO) as our model free algorithm”, however earlier in Section 3 it is mentioned that first the agent is pre-trained with prediction and then the pre-trained model is used with the RL algorithm. Can the authors clarify the training procedure? Is it the case that the Spatial-Net is first pre-trained with some data and then fine-tuned along with the environment rewards? Do the policy-net and the frame prediction net share any parameters?

(b) Is the comparison in Table 2/Figure 5 fair in terms of number of frames seen by the agent? Let a PPO agent see N frames? How many frames does the IPA agent say (both for training spatial Net + Policy).

(c) How about baselines, where instead of augmenting PPO with any additional frames, the Policy is initialized with weights learned by Spatial Net? Other baseline is to jointly optimize for future frame prediction + environment reward (in this case atleast some parameters between the spatial net and the policy net will be shared), but without augmenting the input state with future predicted frames?

The Spatial net architecture is similar to convolutional LSTM — and I therefore don’t think that is a significantly novel technical contribution. The application of spatial net to augment frames in the state is although novel in my best knowledge. The above questions will help me understand the experiments better. Right now the method is slightly unclear to me and the results on ATARI (figure 11) are a bit underwhelming. Also, why did the authors chose the specific ATARI games that they reported results on — why not other games too?

---

> ### Author Response · Authors · 2018-11-19
> **Author Response**
>
> Thank you for the comments. We provide individual responses below.
>
> (a) In our IPA model, the policy net and frame prediction network do not share any parameters. SpatialNet is fine-tuned to more accurately predict future frames on the new environment while the policy net is optimized for control performance. Policy gradients are not back-propagated to SpatialNet. We have added this clarification in the paper (Section 3.2). Both PPO and IPA agents see identical number of frames, we train the frame prediction network on the frames used to train the policy. We have updated the description in Section 3.2 to make this clearer.
>
> (b) All the tested approaches see exactly the same number of frames on the control environment. SpatialNet sees extra frames from the PhysVideos data, but these are offline, from a different domain and contain objects of different shapes, colors, dynamics from the target control environments (PhysWorld and Atari). The PhysVideo frames are only used to train SpatialNet for dynamics prediction, and not for any policy learning.
>
> (c) Thank you for the suggestions. We have added comparisons with the two baselines: (1) ISP: initialization of a model with weights learned with SpatialNet on the PhysWorld environment -- where we use the convolutional encoding of SpatialNet (z_t) as input into the policy network, and  (2) JISP: jointly optimizing future frame prediction + environment reward (see Section 5.1, Table 2).  We find that initializing with the weights from SpatialNet performs about the same as normal PPO, likely due to much of the initially learned priors being corrupted with reward updates. As for the second baseline, we find that joint training does provide a benefit in performance over PPO, but not as large as IPA (except on PhysForage).
>
> (d)  We first emphasize that our experiments are on a stochastic version of Atari, which is more challenging than the benchmark used by previous work [Machado et al., 2017]. Further, we did perform experiments on all Atari games but didn’t present them all due to lack of space. We have added results for the entire suite of Atari games in the appendix in this revised version. We included the Atari results since they are a standard benchmark to compare with previous approaches -- not all games require an understanding of physical dynamics, which explains the cases where our method does not improve upon PPO. We specifically created PhysWorld and performed empirical studies to test our approach on environments that rely more on understanding basic physics like velocity, collision laws, etc.

---

> ### Author Response · Authors · 2018-12-09
> **Request for feedback**
>
> Dear reviewer,
> Thank you so much for your original comments. We spent a large amount of work adjusting the clarifications initially requested. We would appreciate it if you could take a look at the revised version and let us know your thoughts.

---

> > ### Comment · AnonReviewer3 · 2018-12-10
> > **Feedback on Rebuttal**
> >
> > I thank the authors for providing a detailed rebuttal and updating the paper to reflect it. The proposed method outperforms the baselines of ISP and JISP on the physics tasks.
> >
> > My main pro for the paper is that is exhaustively testing the utility of physics prior in learning a goal directed policy. Additional experiments show the proposed method outperforms the closely related I2A baseline.
> >
> > My main concerns are:
> > (a) From figure 11, it seems that IPA and PPO are comparable. While it is true, that IPA outperforms PPO in many more games, but in many cases the performance gain within error bars.
> >
> > (b) The difference between ConvLSTM and Spatial Net is still unclear to me. In one of the comments, authors mention: "SpatialNet also has a input copy mechanism that add the current state to the output of the spatialnet encoding". This is trivial to implement in ConvLSTM -- and will give more insights into why Spatial Net is outperforming ConvLSTM. Right now, results are in favor of Spatial net, but the exact differences from ConvLSTM that lead to performance difference is unclear.
> >
> > Despite a very good rebuttal, I am still concerned by (a)/(b) and can therefore cannot recommend the paper to be accepted. I highly encourage authors to clarify these points and resubmit to a future conference. Utilizing physics prior to improve policy learning has great potential.

---

### Author Response · Authors · 2018-11-19
**Added Requested Baselines, Atari Results, Writing Clarifications**

We thank all the reviewers for the helpful feedback. We have addressed all the comments below as replies to individual reviews. We have also made the following modifications to the revised version of the paper:

a) We have included four additional baselines as suggested by the reviewers including ConvLSTM, imagination augmented agents  (I2A), and two versions of our IPA model: (1) ISP: initialization of a model with weights learned with SpatialNet on the PhysWorld environment -- where we use the convolutional encoding of SpatialNet as input into the policy network, and  (2) JISP: jointly optimizing future frame prediction + environment reward (Section 5.1, Table 2).

b) We emphasize that our experiments are on a stochastic version of Atari, which is more challenging than the benchmark used by previous work [Machado et al., 2017]. We have added results for the entire suite of Atari games (along with standard deviations across runs with different random seeds) in the appendix.

c) We have also added clarifications to the writing suggested by the reviewers, including ego-dynamics (Section 5), descriptions and details of the new baselines (Section 5.1), as well as analysis on their performance compared to our method.

---

### Meta-Review · Area_Chair1 · 2018-12-15

**Confidence:** 4
**Recommendation:** Reject

**Metareview:**

The paper suggests a new way to learn a physics prior, in an action-free way from raw frames. The idea is to "learn the common rules of physics" in some sense (from purely visual observations) and use that as pre-training. The authors made a number of experiments in response to the reviewer concerns, but the submission still fell short of their expectations. In the post-rebuttal discussion, the reviewers mentioned that it's not clear how SpatialNet is different from a ConvLSTM, mentioned the writing quality and the fact that the "physics prior" is really quite close to what others call video prediction in other baselines.